# VideoAlign: A Comprehensive Model for Evaluating Alignment Between Text and Generated Videos

**Yuanming Yang**[1]    **Xiaoqian Liu**[2]    **Jian Chen**[2]

[1]Institute for Interdisciplinary Information Sciences, Tsinghua University
[2]Department of Automation, Tsinghua University
{yym22,lxq21,chenjian20}@mails.tsinghua.edu.cn

## Abstract

Text-to-video generation models have made significant progress recently, but challenges remain in achieving alignment with human preferences. The generated videos frequently lack reliable consistency with their corresponding textual descriptions, and manual evaluation is both labor-intensive and expensive. This study proposes a comprehensive solution to address these alignment issues. We will introduce **VideoAlign**, an end-to-end reward model designed to automatically evaluate the instruction-following capabilities of video generation models.

## 1 Introduction

Text-to-video (T2V) technology, driven by generative models, has seen remarkable advances with the introduction of models like Sora[1], Lumiere[2], StableVideoDiffusion[3] and CogVideoX[4]. These models show great promise in producing high-quality, longer-duration videos that adhere to physical laws.

In T2V research, the instruction-following capability is a key metric, reflecting the consistency between the generated video and its input text. Despite ongoing improvements in video clarity, aligning generated content with textual descriptions remains challenging. Inconsistencies, or even hallucinations, between the video and text can greatly undermine the quality of the output.

Therefore, accurately assessing the instruction-following performance of video generation models is crucial. Given the high cost and limited scalability of manual evaluations, we propose an end-to-end reward model to automate the evaluation of this capability in text-generated videos.

## 2 Related Work

Significant advancements have been made in image understanding and scoring, with models like CLIP[5] and ImageReward[6] contributing notably to image comprehension and evaluation.

However, the temporal dimension in videos, along with substantial changes across frames, complicates the representation of video-text alignment compared to images. While video understanding models, such as dual-stream models[7][8], I3D[9] and CogVLM2-video[10], have made considerable progress in video comprehension and keyframe extraction, studies indicate that general understanding models often fail to align with human preferences in video scoring tasks and struggle with effective automatic alignment scoring (e.g.,TIGER-Lab/GenAI). This discrepancy may arise from the absence of a rigorously defined scoring standard, which requires a well-annotated dataset to guide model scoring. Additionally, the features learned by multimodal understanding models are typically expressed as vocabulary, essentially treating scoring as an ongoing classification task, which may not be optimal in this context.

38th Conference on Neural Information Processing Systems (NeurIPS 2024).

To mitigate this issue, some researchers have attempted supervised learning fine-tuning on well-annotated datasets; however, the evaluation of alignment indicators (e.g., VideoScore–GenAI Tiger-Lab) remains inaccurate.

## 3 Study Proposal

This paper aims to develop a reward model that quantifies the alignment between images, videos, and textual descriptions, facilitating automated scoring.

### 3.1 Datasets Preparation

We will use VidProM[11], a dataset containing extensive T2V pairs from different models.

### 3.2 Base Model

We plan to utilize the latest image and video understanding models, leveraging their robust comprehension capabilities, and perform supervised training on annotated video-text pairs datasets to derive a score that reflects the alignment degree of text-image pairs.

### 3.3 Training Method

We plan to utilize the latest image and video understanding models, leveraging their robust comprehension capabilities, and perform supervised training on annotated video-text pairs datasets to derive a score that reflects the alignment degree of text-image pairs.

While current image and video understanding models are not yet fully mature and may exhibit hallucination issues, our task requires only the generation of a numerical value representing alignment, and we believe these models can effectively accomplish this scoring task.

The proposed reward model not only advances the automated evaluation of multimodal alignment but also establishes a foundation for the future application of reinforcement learning methods, such as PPO and DPO, to improve the trajectory of AI-generated videos.

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
