# OpenReview forum: "VideoAlign: A Comprehensive Model for Evaluating Alignment Between Text and Generated Videos"
_tsinghua.edu.cn/THU/2024/Fall/AML — THU 2024 Fall AML Submission_

### Official Review · ~Juncheng_Yu1 · 2024-11-07
**Exploring Reward Models for Text-to-Video Generation: A Promising Foundation**

**Rating:** 6
**Confidence:** 3

**Review:**

## Summary

This paper proposes a reward model to automatically evaluate the generation quality of text-to-video tasks, which shows the potential to build reward models similar to those used in Reinforcement Learning from Human Feedback (RLHF), laying a foundation for reinforcement learning applications in text-to-video generation.

## Strengths

- **Reasonable Methodology**: Similar to the reward models in RLHF used for training large language models, the proposed method in this paper shows strong potential to automatically generate scores aligned with human preferences.

- **Relevance of Research Problem**: The problem addressed is highly relevant to text-to-video tasks, as it explores the feasibility of constructing reward models to evaluate video generation quality.

## Weaknesses

- **Need for Paper Revision**: Some language issues need to be addressed, such as awkward phrasing (e.g., "studies indicate that general understanding...") and overlooked redundancy in sections (e.g., repeated content in the "Base Model" and "Training Model" sections). It may be beneficial to have co-authors review the paper prior to submission.

- **Lack of Clear Metrics**: The metrics for this reward model are not clearly defined. This lack of clarity leaves the specific goals or target outcomes of the model ambiguous.

- **Potential Insufficiency in Contribution**: The work primarily involves training a scoring head on state-of-the-art (SOTA) models. Expanding the scope of the research with additional experiments or innovations would help enhance the contribution of the paper.

## Score

- **Soundness**: 8/10

- **Contribution**: 6/10

- **Presentation**: 5/10

---

### Official Review · ~Yanchen_Wu1 · 2024-11-08
**Promising research**

**Rating:** 7
**Confidence:** 3

**Review:**

End-to-end design always has many application scenarios, and it is always possible to add a downstream task, using the special structure of the downstream task to conduct the training of the upstream model. This is a very promising research, if it can be done well, it will have a wide range of applications in many content platforms such as auditing, monitoring and recommendation.

It would be easier for me to understand if you could provide some more detailed information about the method.

---

### Official Review · ~Kangping_Xu1 · 2024-11-09
**# Review of "VideoAlign: A Comprehensive Model for Evaluating Alignment Between Text and Generated Videos"**

**Rating:** 7
**Confidence:** 3

**Review:**

## Pros

1. **Timely and Significant Research Direction**
   - Addresses a critical gap in the rapidly evolving field of text-to-video generation
   - The proposed automated evaluation system could significantly accelerate research and development in T2V models by reducing reliance on manual evaluation

2. **Extensible Framework**
   - The proposed reward model could serve as a foundation for future reinforcement learning applications (PPO, DPO)
   - Potential to establish standardized metrics for video-text alignment evaluation across different models and platforms

## Cons

1. **Dataset Limitations and Bias Concerns**
   - The proposal relies solely on the VidProM dataset without addressing its adequacy for training a comprehensive reward model
   - No clear strategy for ensuring sufficient domain coverage and bias mitigation
   - Lacks discussion of potential biases and their mitigation strategies

2. **Unclear Evaluation Methodology**
   - The proposal doesn't specify how the reward model itself will be evaluated for reliability
   - No estimation of human evaluation costs if manual validation is required
   - Absence of clear metrics or benchmarks for assessing the reward model's performance

The proposal presents an important step toward automated evaluation of text-to-video models but requires more detailed consideration of dataset requirements and evaluation methodologies to ensure practical viability.

---

### Official Review · ~Tim_Bakkenes1 · 2024-11-09

**Rating:** 7
**Confidence:** 3

**Review:**

The proposal introduces VideoAlign, an end-to-end reward model designed to automatically evaluate the instruction-following capabilities of text-to-video generation models.

It is a very relevant and interesting topic and you present the background in a way that makes it clear why this is a problem that has to be solved. Being able to automate it in a good way is an innovative approach that would reduce the need for expensive manual evaluation.

You introduce relevant related work and clearly state which dataset you are going to use. However, it would be good to include a bit more ideas on which image and understanding models you plan to use. Also, there does not seem to be a formal problem description with clear metrics in the proposal and the method could be explained a bit more to help the reader understand.

---

### Official Review · ~Lei_Wu17 · 2024-11-09
**Evaluation of the Work: VideoAlign: A Comprehensive Model for Evaluating Alignment Between Text and Generated Videos**

**Rating:** 7
**Confidence:** 4

**Review:**

# Pros:
* Relevance: Addresses a significant gap in T2V evaluation, focusing on alignment accuracy.
* Innovation: Introduces a unique approach with the potential to set a new standard for alignment scoring in T2V.
* Clarity and Structure: Organized and accessible, making complex concepts understandable.
* Scalability: Automation in alignment assessment could dramatically improve evaluation efficiency.
* Research Foundation: Builds on well-established models and datasets, offering a robust theoretical basis.
# Cons:
* Redundancy: Certain sections (e.g., "Training Method") repeat similar points, which could be condensed.
* Potential Model Limitations: As acknowledged, current image and video understanding models may still present challenges in reliability, particularly with hallucination issues.
* Dataset Dependency: Success depends heavily on dataset quality and scope; any bias in the VidProM dataset could affect model accuracy.
* Evaluation Scope: Although thorough, more experimental results or validation data would enhance the assessment of the model's real-world applicability.

---

### Official Review · ~Liutao7 · 2024-11-09
**A research proposal with practical significance**

**Rating:** 7
**Confidence:** 4

**Review:**

The proposal aims to address the alignment issues present in text-to-video generation models, allowing a reward model to automatically assess the model's ability to follow instructions.
Strengths:
Completeness: The proposal is relatively comprehensive, covering aspects such as dataset preparation, base model selection, and training methods, providing a complete framework for the construction of the VideoAlign model.
Creativity: The proposal applies a reward model to the alignment assessment between text and generated videos, which can reduce the human labor cost of video model evaluation.
Workload: The proposal involves manual annotation, training and evaluation of multiple models, and the workload is reasonable.
Suggestions:
Improving the Evaluation System: Consider incorporating discussions or metrics to demonstrate the effectiveness and accuracy of the reward model.
Dataset Issues: The quality of the VidProM dataset directly affects the effectiveness of the reward model.

---

### Official Review · ~Keyu_Shen1 · 2024-11-10
**Well-structured Proposal**

**Rating:** 7
**Confidence:** 3

**Review:**

The proposal presents a promising approach to automate the evaluation of text-to-video (T2V) models, aiming to reduce reliance on labor-intensive manual assessments. Strengths of the proposal include its practical significance and well-defined framework. However, it would benefit from enhancing the evaluation system by incorporating clear metrics to assess the reward model’s effectiveness and addressing potential dataset limitations, as biases in the VidProM dataset could impact accuracy. Additionally, refining redundancy in sections and providing more detailed methodology could strengthen the proposal.

---

### Official Review · ~Guanglei_He1 · 2024-11-11
**This is a very concise proposal.**

**Rating:** 9
**Confidence:** 4

**Review:**

**Pross**

- The approach is clear. Given the inherent difficulty of Text-to-Video (T2V), establishing an effective Reward Model is undoubtedly meaningful for future developments.
- A clearly defined dataset is available for training, which should make it possible to validate the overall effectiveness.

**Cons**

- The specific base model to be used for training the Reward Model is not clearly stated.
- The challenges are significant, as automating the evaluation requires an understanding of the generated video's content, which is itself a complex task.

**Suggestions**

- Consider using a multi-model collaborative approach to build a comprehensive evaluation model. Employ agents to decompose and interpret the generated video, converting it into text, and then compare this text to the original prompt.

---

### Official Review · ~Bowen_Gao1 · 2024-11-12
**Review for VideoAlign: A Comprehensive Model for Evaluating Alignment Between Text and Generated Videos**

**Rating:** 7
**Confidence:** 4

**Review:**

**Summary**

This paper focuses on improving the alignment between prompt text and generated video. The authors propose VidProM as a benchmark dataset and develop a reward model designed to assess alignment scores between images, videos, and textual descriptions.

**Strengths**

1. The paper provides clear background information and a comprehensive overview of related work.
2. The motivation for addressing the alignment issue between text and generated video is well-reasoned.

**Weaknesses**

1. The proposal would benefit from additional figures and mathematical descriptions to better illustrate the approach and provide a clearer, more detailed framework.
2. More information is needed on the specifics of the reward model, including its design and the training process, to enhance understanding of its contribution and effectiveness.
3. Baselines models should be listed.

---

### Official Review · ~Kaiwei_Zhang3 · 2024-11-12
**Well-developed and address a curcial problem.**

**Rating:** 7
**Confidence:** 3

**Review:**

**1. Summary:**

The proposal introduces VideoAlign, an end-to-end reward model designed to evaluate alignment between text prompts and generated videos automatically. VideoAlign aims to address this by using advanced image and video understanding models trained on a large, annotated dataset (VidProM) to score the alignment between text and video. In the future, reinforcement learning techniques, such as PPO and DPO, may be integrated to refine T2V generation based on VideoAlign's scoring.



**2. Overall Quality:**

Overall, the proposal is well-developed, with a promising approach and clear objectives.



**3. Clarity:**

* **Generally clear with specific goal.** The proposal is generally clear, especially in outlining the problem of alignment in text-to-video (T2V) generation and describing the model structure and training approach.
* Current proposal is a little short in length. More details could be provided, especially about baselines.



**4. Originality:**

* **Reward-base model.** Original in focusing on a reward-based model to automate the assessment of T2V alignment, addressing the gap in current T2V evaluations.
* **Longer-duration videos.** While previous work has explored alignment in images or brief video clips, this proposal uniquely targets longer-duration videos and aims for a comprehensive approach.
* **Reinforcement Learning.** Introducing reinforcement learning for further refinement could distinguish VideoAlign in a crowded field.



**5. Significance:**

Address an important problem. It is significant to align videos clips to command text for furthur model development.



**6. Pros:**

* **Address a critical problem.** Without doubt it is crucial to align videos clips to command text in order to enhance the development video-generating models.
* **Utilizes a large dataset (VidProM)**. This could ensure diverse video-text pairs and robust model training.



**7. Cons:**

* **Text repetition.** A paragraph is written twice. *3.2 Base Model* is same to the first paragraph of *3.3 Training Method*.

* **Lack of reference.** In the second paragraph of *2 Related Work*, it is claimed that "general understanding models often fail to align with human preferences in video scoring tasks and struggle with effective automatic alignment scoring". Is this statement supported by any research?

* **Lack of  detailed evaluation metrics and baselines**. How should one evaluate the abilities of the trained reward model? Adding Illustartions to this question in the in the *Study Proposal* section would be nice.

---

### Official Review · ~Fei_Long3 · 2024-11-12
**Innovative Endeavor with Untapped Details**

**Rating:** 7
**Confidence:** 4

**Review:**

**Strength**:

**Novelty and Comprehensive Approach**: The proposal for VideoAlign addresses a significant gap in the field of text-to-video generation by focusing on the alignment between generated videos and their textual descriptions. The authors have outlined a comprehensive model that leverages both image and video understanding models to evaluate the instruction-following capabilities of video generation models. This multimodal approach is a strength as it combines the strengths of different models to tackle the complexity of video-text alignment.


**Weakness**:

**Lack of Rigorous Scoring Standard**: The proposal mentions the absence of a rigorously defined scoring standard but does not offer a solution or a plan to develop one. Establishing such a standard is crucial for the success of the proposed model, and the authors should address this in their future work.

**Lack of Evaluation Metrics and Benchmarks**: The proposal does not clearly outline the specific metrics and benchmarks that will be used to evaluate the performance of the VideoAlign model.

---

### Official Review · ~Yifan_Luo2 · 2024-11-12
**A proposal needs improvement**

**Rating:** 8
**Confidence:** 3

**Review:**

**Summary:**

The document introduces "VideoAlign," a model designed to automatically evaluate the alignment between text and generated videos.

**Pros:**

1. **Automation:** Reduces the need for labor-intensive and expensive manual evaluations.
2. **Scalability:** Facilitates large-scale evaluations, making it easier to test numerous T2V models.

**Cons:**

1. **Model Limitations:** Current video understanding models may still exhibit hallucinations, affecting scoring accuracy.
2. **Data Dependency:** Relies heavily on annotated datasets, which may not cover all scenarios or nuances.
3. **Evaluate Metrics:** How can you evaluate the evaluate model?